# Knee Osteoarthritis Grade Does Not Correlate with Quadriceps Muscle Strength or Bone Properties of the Calcaneus in Men Aged 80 Years or More Who Can Walk Independently

**DOI:** 10.3390/ijerph17051709

**Published:** 2020-03-05

**Authors:** Yoshinori Ishii, Hideo Noguchi, Junko Sato, Hana Ishii, Ryo Ishii, Shin-ichi Toyabe

**Affiliations:** 1Ishii Orthopaedic & Rehabilitation Clinic, 1089 Shimo-Oshi, Gyoda, Saitama 361-0037, Japan; hid_166super@mac.com (H.N.); jun-sato@hotmail.co.jp (J.S.); 2School of Plastic Surgery, Kanazawa Medical University, 1-1 Daigaku Uchinada, Ishikawa 920-0253, Japan; hanamed12@gmail.com; 3Sado General Hospital, 161 Chikusa Sado, Niigata 952-1209, Japan; kmuyakyu@gmail.com; 4Niigata University Crisis Management Office, Niigata University Hospital, Niigata University Graduate School of Medical and Dental Sciences, 1 Asahimachi Dori Niigata, Niigata 951-8520, Japan; toyabe@med.niigata-u.ac.jp

**Keywords:** bone properties of calcaneus, broadband ultrasound attenuation, Kellgren–Lawrence, knee osteoarthritis, quadriceps strength

## Abstract

*Purpose:* Muscle weakness and bone deterioration in the elderly are related to falls and fractures, resulting in decreased mobility. Knee osteoarthritis also may contribute to falls and fractures and thereby affect mortality rates. The Kellgren–Lawrence (KL) classification is widely used in the radiographic evaluation of knee osteoarthritis. *Aims:* This study aimed to evaluate the quadriceps strength and bone properties of the calcaneus for each KL grade, and to clarify the impact of knee osteoarthritis grade on quadriceps strength and bone properties. *Methods*: This prospective cross-sectional study included data on 108 male patients (213 knees), aged ≥80 years, who could walk independently. A handheld dynamometer was used to measure quadriceps strength. Bone properties were evaluated using broadband ultrasound attenuation with a portable bone densitometer. Weight-bearing standing knee radiographs were evaluated using KL classification. Quadriceps strength and bone properties were evaluated for each KL grade and the correlations between the grade and quadriceps strength and bone properties were assessed simultaneously. *Results:* The numbers of participants in KL grades I–IV were 46, 102, 45, and 20, respectively. There were no differences among grades for either quadriceps strength or bone properties. *Conclusions*: Participants exhibited good quadriceps strength and bone properties regardless of their KL grade. Relatively high mechanical loading of muscle and bone incurred while walking independently, likely explaining this result. Clinically, this study demonstrated the absence of correlations between KL grade and quadriceps strength and bone properties, as was previously reported in studies showing the absence of a correlation between KL grade and pain.

## 1. Introduction

Patients with osteoarthritis and walking disabilities are at a higher risk of death than the general population [1]. Consequently, effective management of these patients may be important to reduce mortality. The Kellgren–Lawrence (KL) classification [2] is commonly used as a research tool in epidemiological studies of osteoarthritis [3,4] and as an indicator of the incidence or progression of knee osteoarthritis [3,4]. Previous studies on clinical symptoms (such as muscle strength around the knee [3,4,5,6,7]) and bone properties (using bone mineral density (BMD) [8,9]) associated with each KL grade have been reported. However, there is currently no consensus on the association of KL grade with muscle strength and bone properties in patients with knee osteoarthritis.

The decline in quadriceps muscle strength (QS) in older adults is reportedly associated with cognitive impairment [10]. Moreover, deterioration of bone properties (as evaluated by BMD) is the primary factor related to femoral neck fractures [11]. Both cognitive impairment and fractures lead to a deterioration in mobility [12,13]. Therefore, maintaining effective levels of muscle strength and bone properties seems to be important for preserving the mobility of older people.

Currently, only 11 countries have an average male life expectancy of 80 years or more; much fewer than for women [14]. To the best of our knowledge, there are no reports that focus on men aged more than 80 years with various KL grades who can walk independently and analyzed how their level of independent walking affects their muscle strength and bone properties. Establishing this information may help to provide target values to facilitate the maintenance of independent walking in older adults who have not yet reached the age of 80 years. Furthermore, taking the significant correlation between walking level and mortality [15] into account, achieving or exceeding these values may prolong the healthy life span (i.e., the period during which there is no restriction on daily life caused by mobility limitations).

A portable handheld dynamometer is often used to measure muscle strength because reliable values can be obtained [3,4,16,17]. Omori et al. [3] reported a significant correlation between muscle strength measurements done with a portable dynamometer and with a large-scale machine (Biodex System 3, BDX-3; Biodex Medical System Inc., Shirley, NY, USA). A handheld dynamometer is small, lightweight, and portable and can be used for quantitative evaluation of muscle strength even at small-scale facilities. Moreover, bone properties obtained with a portable bone densitometry system that uses calcaneus broadband ultrasound attenuation (BUA; dB/MHz) have been reported. The use of BUA is less expensive than dual-energy X-ray absorptiometry (DEXA) and is radiation free [18,19,20,21,22,23]. Several authors [18,20] have reported significant correlations between BUA values and BMD values obtained using DEXA. 

The purpose of this study was to evaluate both QS and the bone properties of the calcaneus (BPC) in men over 80 years of age who could walk independently and to clarify whether their KL grade correlated with QS and BPC. The hypothesis of this study was that men aged more than 80 years who can walk independently do not show a significant correlation between KL grade and QS and BPC because of the maintenance of mechanical loading on the body as a result of independent walking.

## 2. Materials and Methods

### 2.1. Participants

This prospective cross-sectional study was conducted from September 2018 to August 2019. The study included data on 108 Japanese male patients (213 knees) who visited an orthopedic specialty outpatient clinic (median age: 83 years, 25th percentile: 81 years, 75th percentile: 86 years, range: 80–94 years). Their median (25th percentile, 75th percentile) body weight (BW) (kg) was 59 (56, 66) and their median height (cm) was 159 (156,164). The median body mass index was 24 (22, 25). Three knees were excluded because they had undergone total knee arthroplasty (TKA). All 108 patients received diagnoses of degenerative joint and/or cartilage disease. More specifically, the diagnoses were spine-related for 52 patients, upper limb-related for 14 patients, lower limb-related for 40 patients, and related to other areas for two patients. To be included in this study, patients had to be men aged 80 years or older who were able to walk independently with or without a T-cane but without support from other people. The exclusion criteria were as follows: (1) neurological findings, such as motor paralysis; (2) cognitive or mental dysfunction requiring medication; and (3) skeletal dysfunction that had a negative impact on walking. This study was conducted based on the guidelines laid down in the Helsinki Declaration and in the ethical guidelines of our institution. The Research Board of Healthcare Corporation Ashinokai, Gyoda, Saitama, Japan, approved the study (ID number: 2018-6). All patients provided written informed consent for participation.

### 2.2. Muscle Strength and Bone Property Measurements

#### 2.2.1. Quadriceps Muscle Strength

To assess QS, the isometric knee extension muscle strength in Newtons was measured with the knee in approximately 20° of flexion using a Locomo Scan dynamometer (Alcare Corp., Tokyo, Japan), following a standard protocol [3]. The correlation between the maximum measured value using this arthrometer and the value obtained using the Biodex System 3 was r = 0.69 (p < 0.01) for the first measurement and r = 0.82 (p < 0.01) for the second measurement conducted 1 week later. Moreover, a good correlation (r = 0.92, p < 0.01) was also observed between the first and second measurements obtained using the Locomo Scan arthrometer [3]. Two measurements of QS were taken, and the highest value on each side was used in the analysis. These values were divided by BW to eliminate the effect of physique. Both the absolute QS values and the normalized QS values (i.e., the QS/BW ratios; N/kg) were used in the analysis. 

#### 2.2.2. Bone Properties of the Calcaneus

BPC was assessed using bilateral calcaneus BUA readings obtained with an AOS-100SA ultrasound system (Hitachi-Aloka Medical, Ltd, Tokyo, Japan). All measurements were taken with the same device. Two transducers (receiving and emitting) faced with rubber coupling pads were placed in direct contact with either side of the patient’s heel. Ultrasound gel was applied to the coupling pads to ensure good contact. Standard protocols for this instrument were used to obtain the BUA, including calibration and positioning of the participants. After measurement of the absolute BPC values, they were divided by BW to eliminate the effect of physique. Both the absolute BPC values and the resulting BPC/BW ratios were used in the analysis. The intraclass correlation coefficient for calcaneus BUA, calculated to assess test–retest reliability, was 0.988 (95% confidence interval: 0.955–0.997).

### 2.3. Radiographic Assessment

All participants underwent a Rosenberg-view anteroposterior (AP) radiographic examination of their knees during weight-bearing [24]. The KL grade was defined using the KL radiographic atlas for overall knee radiographic grades [2]. Radiographs were scored from KL grade 0 to grade IV, with higher grades being associated with more severe osteoarthritis. Each grade was defined using AP radiographs as follows: grade I, doubtful narrowing of the joint space with possible osteophyte formation; grade II, possible narrowing of the joint space with definite osteophyte formation; grade III, definite narrowing of joint space, moderate osteophyte formation, some sclerosis, and possible deformity of bone ends; and grade IV, large osteophyte formation, severe narrowing of the joint space with marked sclerosis, and definite deformity of bone ends. To evaluate the intra-observer variability of KL grading, 100 randomly selected knee radiographs were scored by the same observer [YI] more than one month after the first reading. The intra-observer variability evaluated for the KL grade (1 to 4) was confirmed by kappa analysis to be sufficient for assessment (k = 0.894). 

QS and BPC values for men aged 80 years or more who could walk independently were measured for each KL grade, and the correlation of the KL grade with QS and BPC values were evaluated simultaneously. 

### 2.4. Statistical Analysis

Data for some variables did not pass the Kolmogorov–Smirnov test for normality. The effects of the KL grade on QS and BPC were analyzed using the Kruskal–Wallis method. If necessary, Bonferroni’s method was used to adjust for multiple comparisons between QS and BPC values at respective KL grades. In all tests, a *p* value < 0.05 was considered statistically significant. Post hoc power analysis was performed to evaluate whether the data had sufficient verification power. The estimated power was 0.88, with an effect size of 0.25 and an alpha error of 0.05. All statistical analyses were performed using IBM SPSS Statistics, Version 23 (IBM Japan, Tokyo, Japan). 

## 3. Results

The numbers of participants stratified into the four categories according to the KL grade were as follows: grade I, 46; grade II, 102; grade III, 45; and grade IV, 20. Among the participants’ background parameters, only BMI showed a significant difference among the four groups using the Kruskal–Wallis test (*p* = 0.043) (Table 1). However, no differences were found between the four grades with multiple comparisons using Bonferroni’s method. Moreover, there were no differences among grades for either the absolute or weight-adjusted values for QS and BPC (Table 1).

## 4. Discussion

The most important findings in this study were that there was no correlation between the KL grade and muscle strength (QS, QS/BW) and between KL and bone properties (BPC, BPC/BW) in men aged 80 years or more who were able to walk independently. These findings clarified that radiographic knee osteoarthritis evaluation does not necessarily reflect QS or BPC. Moreover, the index values of QS and BPC established in the current study may indicate target values to enable the elderly to prolong independent walking up to and beyond the age of 80.

Considering the correlation between knee osteoarthritis and muscle strength, a decline in quadriceps strength was reportedly related to the incidence of osteoarthritis [3,4,5]. However, several studies have published findings inconsistent with the above-mentioned correlation between knee osteoarthritis and muscle weakness. In a systematic review and meta-analysis, Øiestad et al. [25] reported that knee extensor muscle weakness was associated with an increased risk of developing knee osteoarthritis in both men and women. In contrast, in a longitudinal study, Takagi et al. [4] recently showed that quadriceps muscle weakness is related to an increased risk of radiographic knee osteoarthritis, but not to its progression. Furthermore, several authors [5,6] reported a gender difference between deterioration of osteoarthritis grade and muscle weakness. Omori et al. [3] showed that the tendency for muscle strength to decline with the progression of knee osteoarthritis grade was particularly evident between KL grade 0 and grade I in both men and women, and between KL grade I and grade II in men. Conversely, Palmieri-Smith et al. [26] did not find any differences in muscle strength among KL grades, and Ruhdorfer et al. [7] reported no direct association between muscle strength and KL severity.

In contrast to some of the above findings, muscle strength has been found to be affected not by KL osteoarthritis grade but by pain [7]. Studies investigating the correlation between osteoarthritis grade and pain found that radiographic knee osteoarthritis did not necessarily reflect pain [27,28]. Additionally, increasing age is protective for reporting pain [28,29]. Finally, the current study showed no difference in QS between KL grades. As a result, it is reasonable to speculate that our participants, who were aged 80 years or more and could walk unaided, might have little or no involvement of knee pain impacting on QS, despite their KL grade. 

A few recent studies [16,17] evaluated QS using the same arthrometer as that used in the current study. Ishii et al. [16] reported that the median QS/BW ratio was 3.3 kgf/kg for patients who had undergone posterior cruciate ligament-retaining TKA, 3.4 kgf/kg for patients with posterior cruciate ligament-substituting TKA (median age: 81 years), and 4.6 kgf/kg for the controls (median age: 83 years). For Japanese men aged 80–89 years (n = 86), Narumi et al. [17] reported the median QS/BW ratio to be 5.9 kgf/kg. The values for QS found in the present study were mostly consistent with these previous findings. We believe that the index values of QS reported in the present study can be used as target values for adults aged less than 80 years who wish to maintain independent walking ability up to and beyond age 80 years.

Several studies [21,22,23] have demonstrated significantly improved BUA values as a result of the increased mechanical loading to the calcaneus, due to walking [22], weight-bearing exercise [23], and pain relief after TKA [21]. Ishii et al. [19] compared the bone quality of patients who underwent TKA, age-matched controls, and patients who suffered from hip fracture. The average BUA values of the TKA group (*n* = 70, average age 76 years), controls (*n* = 70, average age 76 years), and the hip fracture group (*n* = 107, average age 83 years) were 42.2 ± 15.5, 40.2 ± 14.9, and 22.9 ± 13.3 dB/MHz, respectively. The BUA values obtained in the current study were higher than these values. Therefore, it seems reasonable to suggest that participants who can walk independently may maintain good bone properties because of their higher activity levels regardless of their radiographic osteoarthritis grade.

Measurements of BMD are much more commonly reported to evaluate bone properties in the elderly than BUA is. The reason for the more common use of BMD than BUA is that BMD measurements enable the quantitative evaluation of variations in bone biomechanical characteristics such as the trabecular bone strength [30]. However, the relationship between radiographic osteoarthritis grade and BMD is still inconclusive. One report [9] indicated that BMD was lower with severe knee osteoarthritis before TKA surgery, whereas another [8] found that BMD was not related to the severity of knee osteoarthritis in postmenopausal women. Graafmans et al. [18] investigated the relationship between BMD and BUA and found that BUA at the heel can predict BMD at the hip. Ishii et al. [20] indicated that BUA in the calcaneus correlated significantly with BMD at the proximal femur and tibia. Their report stated that the noninvasive BUA score may predict BMD at these locations in the 10-year TKA follow-up study. Consequently, although BMDs were not measured in the current study, it is reasonable to speculate that the values for proximal hip and tibial BMD are likely to show age-appropriate values or even higher values, given the high BUA values of the current participants.

This study has several limitations. First, the participants suffered from some skeletal dysfunction caused by degenerative joint and/or cartilage disease. However, because of the inclusion criterion that participants did not require the support of others to walk, the impact of their disease status and their various background parameters on QS and BPC may have been negligible or small. Second, we evaluated radiographic knee osteoarthritis only. Currently, we are conducting additional investigations including symptomatic evaluation, using the total West Ontario and McMaster University Osteoarthritis Index (WOMAC) score, and quantitative mobility, such as steps and walking distance, using a pedometer. Third, we did not consider other factors that may impact the QS and BPC, such as age, sex, and ethnicity. Finally, this was a single-center study, which may limit the generalizability of the results. Verification of the validity of our results through research at multiple facilities is expected in the future.

Despite these limitations, the findings of the current study are valuable because they show that even elderly subjects aged 80 years or more could maintain good muscle strength and bone properties regardless of their radiographic osteoarthritis grade, provided that they can walk independently. The overall finding of this study was the absence of a correlation between body factors, such as QS and BPC, and radiographic osteoarthritis. This finding was consistent with previously reported discordance between pain and radiographic osteoarthritis [27,28]. The reference ranges established in the present study can be used by older adults who have not yet reached the age of 80 years and their health care providers as target values to facilitate the maintenance of independent walking. 

## 5. Conclusions

Men aged 80 years or more who could walk independently were able to maintain QS and BPC regardless of the degree of radiographic knee osteoarthritis. In the future, to prolong the healthy life span of the elderly, we believe that further studies will be necessary to investigate the effects of the degree of radiographic knee osteoarthritis on lower limb function parameters such as walking speed, the timed up and go test, one-legged stance, and the functional reach test as well as the body factors analyzed in this study.

## Figures and Tables

**Table 1 ijerph-17-01709-t001:** Isometric quadriceps strength and bone quality of the calcaneus for men aged 80 years or older (*n* = 108 patients, 213 knees).

K-L Grade (Knees)	Grade I (46)	Grade II (102)	Grade III (45)	Grade IV (20)	*p*
Age (year)	82[81, 85]	84[81, 86]	83[82, 86]	85[82, 86]	0.195
Body weight (kg)	57[55, 66]	59[55, 66]	61[58, 64]	62[56, 67]	0.399
Body height (cm)	158[156, 162]	160[156, 164]	157[155, 163]	159[152, 164]	0.457
BMI (kg/cm^2^)	23[22, 25]	23[21, 25]	24[22, 26]	25[23, 26]	0.043
Quadriceps strength					
Absolute values (N)	303[214, 361] (82–623)	311[241, 385] (97–679)	299[218, 363] (91–584)	302[266, 324] (147–495)	0.596
weight-adjusted values (N/kg)	4.8[3.9,5.8] (2.0–9.9)	5.3[4.3,6.3] (1.6–10.1)	4.8[3.6,6.4] (1.6–10.1)	4.8[3.6-5.7] (2.6–8.2)	0.421
Bone Quality (BUA; dB/MHz)					
Absolute values (dB/MHz)	55[44, 65] (19–88)	56[47, 64] (23–109)	60[43, 68] (17–92)	58[52–65] (16–85)	0.800
weight-adjusted values (dB/MHz /kg)	1.0[0.7,1.1] (0.2–1.6)	1.0[0.8,1.1] (0.5–2.0)	1.0[0.8-1.1] (0.3–1.5)	0.9[0.7-1.1] (0.3–1.6)	0.971

K-L; Kellgren–Lawrence. BUA; broadband ultrasound analysis. Values are presented as medians [interquartile range] (range).

## Data Availability

The datasets used and/or analyzed during the current study are available from the corresponding author on reasonable request.

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
