# Peer review of "Knee Osteoarthritis Grade Does Not Correlate with Quadriceps Muscle Strength or Bone Properties of the Calcaneus in Men Aged 80 Years or More Who Can Walk Independently"

_ijerph, 2020, doi:10.3390/ijerph17051709_

Round 1

Reviewer 1 Report

Review for Manuscript ijerph-715853-peer-review-v1

General Comments: Very succinct, nicely written manuscript. The addition of a post-hoc power analysis provides further strength to the conclusions since the study was adequately powered. One major comment – While the overall evaluation of the data for BMI reached statistical significance, none of the post-hoc pair wise analyses reached statistical significance between groups. Perhaps report the lowest (closest to significance) post-hoc P values and indicate which groups are compared? The data shows that Grade I and II had the lowest BMIs.

More Specific Comments:

Title – None

Abstract

Line 33 – Insert a comma after “independently” Line 34 – Change “explains” to “explaining” Line 35-36 – Clarify if this previous study referenced did or did not indicate a correlation.

Introduction – None

Materials and Methods

Line 137 – Insert “for normality” after “test” Line 140 – As mentioned, nice inclusion of the post-hoc power analysis.

Results – See above comment about potentially reporting the post-hoc P values for BMI.

Discussion – None

Conclusions – None

Figure and Table Legends – None

Figures and Tables – None

Author Response

Our manuscript underwent professional editing by David Smallbones, BSc from Edanz Group (www.edanzediting.com.ac).

Reviewer 2 Report

  1. Were the subjects doing any form of exercise? Were there any significant differences in the activity levels of these subjects? 
  2. Were there any exclusion criteria for BMI? The range of the BMI is very narrow and includes very fit people. 
  3. Were there any specific times at which the measurements were taken? Day? evening? Was it always the same for all the subjects and all the measurements? 
  4. Why were any pain measurements not included in the study? 
  5. Do the authors think that the paper would have benefitted from including correlation graphs in the results section?
  6. Was it possible to include control subjects in the study to determine if the QS and BPC values showed differences between non-OA and OA subjects?

Author Response

Our manuscript underwent professional editing by David Smallbones, BSc from Edanz Group (www.edanzediting.com.ac.).
